# Dental Students’ Experiences during the COVID-19 Pandemic—A Cross-Sectional Study from Norway

**DOI:** 10.3390/ijerph19053102

**Published:** 2022-03-06

**Authors:** Ida Heitmann Løset, Torgils Lægreid, Ewa Rodakowska

**Affiliations:** Department of Clinical Dentistry, Section of Cariology, University of Bergen, 5009 Bergen, Norway; ida.loset@student.uib.no (I.H.L.); torgils.lagreid@uib.no (T.L.)

**Keywords:** COVID-19, pandemic, stress, dental students, education

## Abstract

The purpose of this cross-sectional study was to map dental students’ experience of the study situation throughout the pandemic. All clinical dental students (year 3 to 5) at the Faculty of Medicine, Department of Clinical Dentistry (IKO), University of Bergen (UiB), Norway, were invited. Participation was anonymous and voluntary, and the response rate was 63%. Questions regarding stress-related factors were divided into three categories. In the category «Stressors/learning», a statistically significant difference was observed between both the genders (*p* = 0.001) and years of study (*p* = 0.028). Statistically significant differences between the genders were also observed in the category «Stressors/infection» (*p* = 0.008). Women were significantly more stressed due to lack of clinical skills (*p* = 0.048), not receiving as good theoretical teaching as before the pandemic (*p* = 0.016), and uncertain issues around the exams (*p* = 0.000). Fourth year students were significantly more stressed due to lack of clinical skills (*p* = 0.012), for not passing the clinic/skills courses due to lack of study progression (*p* = 0.005), and worries about not being a good enough dentist after graduation (*p* = 0.002). In conclusion, the pandemic had a major impact on dental students. The most prominent stressors in relation to the study situation were experienced by students from the fourth year and female students. Clinical and theoretical learning outcomes among students were regarded as worse than before the pandemic. The students preferred in presence lectures, but experienced digital asynchronous video lectures as a good alternative. The pandemic negatively affected the students’ social life. Dental schools should be aware that students have been exposed to increased distress and burden through the pandemic and should provide support for those in need.

## 1. Introduction

In the beginning of March 2020, the World Health Organization (WHO) declared the outbreak of COVID-19 as a pandemic [1]. Nobody expected that our professional and private life would change so significantly. The first applied restrictions in many countries aimed to slow down the spread of infection and to avoid overloading the capacity of the health care system. Universities, among other institutions and businesses, were closed down, and all traditional teaching had to be quickly converted into digital education.

Dental education is demanding, with a major part of practical training in addition to theoretical teaching. The dental students spend each year a lot of time with face-to-face teaching and hands-on practice to develop their manual dexterity and clinical skills. Dental students in Norway are among those students who spend the most time on their studies under normal circumstances (an average of 47–48 h per week) [2]. With the COVID-19 pandemic and the limitations entailed, the dental students lost a lot of time in the clinic and thus much hands-on training. Patient treatment was partially replaced by digital clinical seminars and assignments. Lectures were conducted on digital platforms, and there were changes in exam forms. To carry out scheduled activities, some semesters were extended at the expense of holidays.

At the beginning of the pandemic, a special infection control committee was established at the Department of Clinical Dentistry at the University of Bergen (UiB) to introduce reinforced infection control measures. These measures were continuously adjusted during the pandemic based on changes in the pandemic situation in Norway and the at any time applicable national guidelines. Reinforced infection control measures included no tolerance for attendance in the clinic with symptoms of respiratory infection for all patients, students, and staff. There was a general requirement for the use of face masks in all the department’s premises and good hand hygiene, in addition to the use of extra infection control equipment during patient treatment.

Studies show that dental students in a normal setting experience much stress throughout their education [3,4,5,6,7]. The main sources of stress include workload, clinical requirements, assessments, and grades [3,4,5,6,7,8]. In the clinical part of the education, students need to perform treatment of patients to achieve good clinical skills and necessary study progression. At the dental educations in Norway, there are minimum clinical requirements that the students must meet to pass the clinical service. These stressors and demands can affect dental students’ academic performance, psychological well-being, and physical health [3,9]. Many studies published globally before the pandemic, showed a high incidence of depression, anxiety, and stress among dental students [3,4,5,6,7].

The purpose of this study was to map dental students’ experience of the study situation throughout the pandemic. We wanted to take a closer look at stressors present in the study situation during the pandemic and the students’ learning experience to be better prepared for similar situations and draw conclusions for the future.

## 2. Materials and Methods

### 2.1. Study Design and Participants

This was a cross-sectional study carried out between 17 March and 12 April in 2021. It was approved by the Regional Committee for Medical and Health Research Ethics (REK) (reference number 230726) in conformity with the Helsinki Declaration guidelines. All dental students at the Faculty of Medicine, Department of Clinical Dentistry (IKO), University of Bergen (UiB), Norway, who had started their clinical training (year 3 to 5) were invited to participate in the study (*n* = 134). Participation was both anonymous and voluntary. All the participants signed an informed consent prior to the access to the questionnaire. On the 17th of March 2021, an invitation was distributed via the digital learning platform MittUiB (Canvas) with a link to the questionnaire. It assured anonymity of all data.

### 2.2. Questionnaire

The questionnaire was designed specifically for this study. It contained questions about gender and year of study, as well as sixteen questions that dealt with stress-related factors, six questions about learning experience, two questions about consequences, and two questions about social factors. Altogether, the questionnaire consisted of 28 questions, including 14 sub-questions. They were closed-ended in the form of multiple choice or Likert scale. The respondent could not proceed with the questionnaire until these questions had been answered. One question was open-ended and voluntary to answer. Our questionnaire was organized electronically with the use of the software SurveyXAct (Rambøll Management Consulting, Oslo, Norway). As a pilot, the questionnaire was tested on a group of volunteer students (*n* = 25). The participants were asked to give feedback on the questionnaire and after minor corrections, we prepared the final version. The answers to the open-ended question were presented graphically with the “Pro Word Cloud” (Orpheus Technology Ltd.: London, UK) extension in Microsoft^®^ Word.

### 2.3. Statistical Analysis

The statistical analysis was performed using the IBM SPSS Statistics, Version 26.0 (IBM Corp., Armonk, NY, USA). A two-group t-test was used to examine mean differences between women and men, whereas a one-way ANOVA test was used to examine mean differences among third, fourth and fifth year students. Residuals were inspected to see if these followed a bell-shape in order that the mean value was sufficiently robust estimated. The significance level was set at 5% (*p* < 0.05).

## 3. Results

The response rate was 63%. More women participated in the study and fourth year students were most represented. The demographic characteristics are presented in Table 1.

Table 2 presents questions regarding stress-related factors. Over 80% of the students reported that they had perceived more stress due to the study situation during the pandemic than before, but there were no significant differences between the genders or the years of study. Questions were divided into the following categories: «Stressors/learning», «Stressors/infection» and «Stressors/consequences». The sum score in the category “Stressors/learning” showed significant differences between both the genders (*p* = 0.001) and the years of study (*p* = 0.028). Moreover, significant difference between the genders was observed in the category «Stressors/infection» (*p* = 0.008). Women were significantly more stressed due to lack of clinical skills (*p* = 0.048), not receiving as good theoretical teaching as before the pandemic (*p* = 0.016) and uncertain issues around the exams (*p* = 0.000). In addition, women reported significantly more stress affecting their concentration (*p* = 0.019). Students in the fourth year were significantly more stressed due to lack of clinical skills (*p* = 0.012), for not passing the clinic/skills courses due to lack of study progression (*p* = 0.005), and worries about not being a good enough dentist after graduation (*p* = 0.002).

Another finding in this study was that most of the students preferred to go back to in presence (physical) lectures when the pandemic is over (Figure 1), followed by digital asynchronous video lectures.

Learning experience through the pandemic are presented in Table 3. 60% of the students reported that the theoretical learning outcomes had been either worse or much worse during the pandemic, but there was no significant difference between the genders or the years of study. The male students in the fourth and fifth years experienced significantly poorer clinical learning outcomes than the females in the fourth and fifth years (*p* = 0.027).

The pandemic has negatively affected the students’ social life, but there was no significant difference between the genders or the years of study. One in five students had considered quitting or applying for leave from their studies due to reasons related to the pandemic, or already did. The reasons mentioned by the students were uncertainty and stress because of the pandemic, little clinical training, a lot of absence due to quarantine/isolation, lack of social contact, and mental health.

Figure 2 presents a word cloud for all comments on the open question. The word cloud is an overview of all keywords in the answers. All single words in the students’ answers were summarized, and the most repetitive words are presented with the largest font size. Factors related to the clinical teaching were the most prominent stressors among the students.

## 4. Discussion

Medical, dental, and healthcare education are in general regarded as emotionally and academically demanding, and the prevalence of anxiety and depression is higher than in the general population [3,9,10,11]. According to a meta-analysis of Quek et al., the prevalence of anxiety among medical students was 33.8%, whereas in the general population, did not exceed 25% depending on different surveys [10,12,13]. Moreover, Guse et al. showed that dental students were even more distressed than medical students during the pandemic [9]. Many factors influence this situation, among others the workload. The COVID-19 pandemic had a significant impact on our private and professional life, including education. Moreover, the pandemic situation was an additional stressor for the students. To our knowledge, this was the first study in Norway during the pandemic that focused on dental students’ experiences of stressors related to the demanding study situation. The main findings of our study are the impact of the pandemic on fourth-year students and female students.

The strength of this study is the relatively high response rate, especially to the open-ended question (77%). This may indicate a clear engagement in the topic, and that it was important for the respondents to share thoughts. Another strength was that the questionnaire was produced specifically for this study. A frequently used questionnaire in other studies on dental students is the Dental Environmental Stress (DES) Questionnaire [3,4,14]. In our opinion, this questionnaire did not fit into the purpose of our study. On the other hand, this can cause a limitation because the findings are unique only for this sample and cannot be generalized to all dental schools and dental educations.

Most students in the present study had experienced more distress and burden during the pandemic, and this was mostly related to clinical teaching. Patient care is essential for developing technical skills required in the dental profession and this kind of teaching is difficult to digitize since it involves treatment of patients. In addition, the development of communicative abilities and empathy is another crucial part of the education that cannot easily be replaced by digital aids. Although there is a potential in the use of digital tools in dental/medical education, students are not always satisfied with this kind of teaching [15,16]. Even if students highly accept remote teaching their manual dexterity is not developing [17], and digital devices must be considered as a supplement [18,19].

The fourth year students experienced the most burdening learning situation. They were more concerned about lack of clinical skills compared with students in the third and fifth year. Students of the fourth year usually receive a considerable amount of clinical training during the academic year, but the pandemic situation affected this significantly. However, results from other dental educations in Europe showed no significant differences in terms of clinical factors between students in the clinical years [20,21]. On the other hand, according to Zarzecka et al., fifth year students experienced significantly more stress than the second to fourth year students regarding «academic work» [21]. In general, it appears that students in the clinical part of the education experience more stress than students in the preclinical years [4,20,22]. Moreover, fourth year students were also concerned about proper study progression. Lack of study progression may have financial consequences. Retaking one year can lead to lost income and increased student loans. Furthermore, students from the fourth year were worried about their professional future. They were worried about not being sufficiently trained as dentists when they graduate. This may be related to the fact that they had limited access to treat patients and, due to the pandemic, patients were cancelling their appointments.

Studies show that men and women respond differently to the experience of stress [23,24,25,26]. Females often respond more emotionally to stress, whereas men tend to respond with aggression to severe stress [23,27,28,29]. Moreover, cultural norms may influence the experience of stress [27,28]. According to the literature, women are generally more prone to stress, anxiety, and depression than men [28,30]. In the Students’ Health and Well-being Survey (SHoT) [31] from 2021, 46% of the students on a national basis stated that they experienced stress as a health problem. The same survey also showed a significant increase in mental illness since the previous survey in 2018 (45% vs. 32%). The increase was most pronounced among female students [31]. Mekhemar et al. showed that dental students experienced up to mild anxiety, stress, and depression during the pandemic, and women were significantly more affected than men [32]. Nevertheless, there are studies showing that male students were affected significantly more by stressors [33], whereas others are showing no statistical significance between gender regarding changes of mental health and worries about study situation and motivation [9].

Additionally, our study showed that women were more stressed than men regarding stressors in the learning situation. This was consistent with Agius et al. [20], who showed that female dental students experienced more concerns about exam changes and loss of manual skills than the male students during the pandemic. Other studies point out the loss of manual skills as the most prominent stressor among dental students [20,34]. Furthermore, exam-related stress is also among the biggest stressors under normal circumstances [4].

Most students in the present study wanted to return to in presence lectures after the pandemic. At the same time, a large proportion of the students thought that asynchronous video lectures were a good alternative to traditional lectures. On the other hand, synchronous lectures on., e.g., Zoom were less desirable among the students, and were consistent with results from another survey [35]. A noteworthy finding was that almost two out of three students were disappointed with the theoretical learning outcome compared with before the pandemic. This is in contrast to the results from Bunæs et al. earlier in the pandemic [36]. Digital teaching seems to be less engaging, and the students miss the contact with their fellow students [31]. This finding could also be related to impaired communication with the school [2,21,24].

At the same time, students’ social life was negatively affected during the pandemic due to distance requirements, closed social meeting places, and gyms. Uncertainty and insecurity related to pandemic may be the reasons for quitting or applying for leave from studies.

## 5. Conclusions

The pandemic had a major impact on dental students. The most pronounced stressors in relation to the study situation were experienced by students from the fourth year and female students. Clinical and theoretical learning outcomes among students were regarded as worse than before the pandemic. The students preferred in presence lectures, but experienced digital asynchronous video lectures as a good alternative. The pandemic negatively affected the students’ social life. Dental schools should be aware that students have been exposed to increased distress and burden through the pandemic and should provide support for those who are in need.

## Figures and Tables

**Figure 1 ijerph-19-03102-f001:**
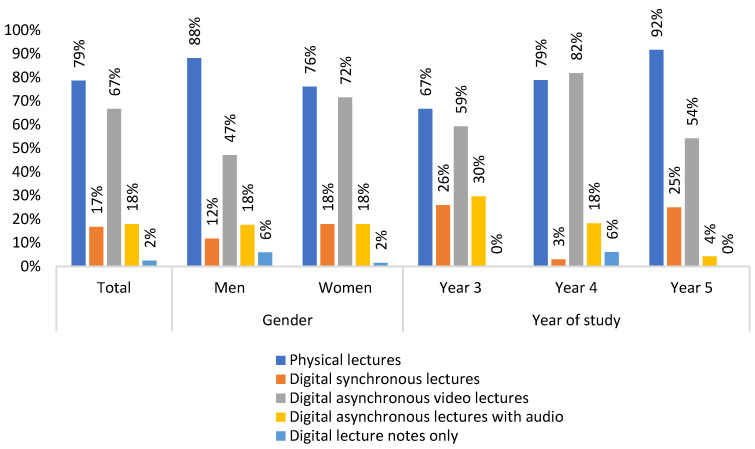
Preferred methods of theoretical teaching after the pandemic.

**Figure 2 ijerph-19-03102-f002:**
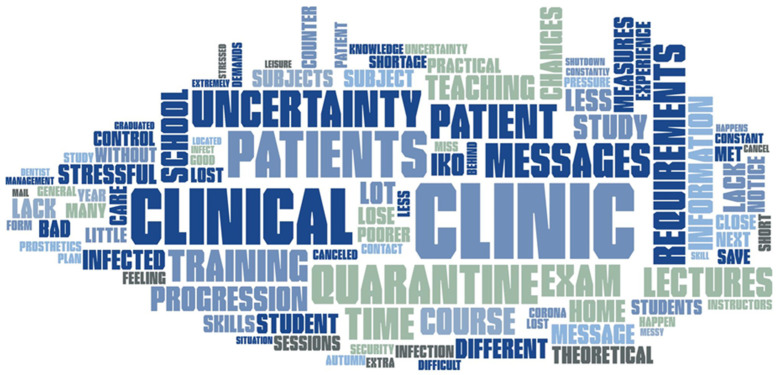
Word cloud as an overview to open question answers.

**Table 1 ijerph-19-03102-t001:** Demographic and response characteristics of the participants.

		Total Number of Students (*n*)	Number of Attended Participants (*n*)	Response Rate
		134	84	63%
Gender	Men	41	17	41%
	Women	93	67	72%
Year of study	Year 3	46	27	59%
	Year 4	48	33	69%
	Year 5	40	24	60%

**Table 2 ijerph-19-03102-t002:** Stress-related factors.

Stress Due to:	Answers in Likert Scale			Mean ± SD		*p*-Value Gender	*p*-Value Year of Study
	1 + 2	3	4 + 5				
The study situation during the pandemic	7.2%	8.4%	83.4%	Men	3.9 ± 0.9	0.218	0.123
				Women	4.2 ± 0.9		
				Year 3	3.9 ± 1.0		
				Year 4	4.4 ± 0.7		
				Year 5	4.0 ± 0.9		
Category «Stressors/learning»				Men	18.3 ± 5.0	0.001 *	0.028 **
				Women	22.5 ± 4.4		
				Year 3	21.0 ± 5.5		
				Year 4	23.3 ± 4.5		
				Year 5	20.0 ± 3.8		
Lack of clinical skills as a result of lost clinical/skills training	11.9%	17.9%	70.2%	Men	3.4 ± 1.3	0.048 *	0.012 **
				Women	4.0 ± 1.0		
				Year 3	3.7 ± 1.1		
				Year 4	4.3 ± 1.0		
				Year 5	3.5 ± 1.1		
Not passing the clinic/skills course due to lack of study progression	18.0%	21.7%	60.3%	Men	3.4 ± 1.0	0.361	0.005 **
				Women	3.7 ± 1.3		
				Year 3	3.8 ± 1.3		
				Year 4	4.1 ± 0.9		
				Year 5	3.0 ± 1.4		
Not receiving as good theoretical teaching as before the pandemic, due to changed teaching methods	27.4%	27.4%	45.2%	Men	2.6 ± 1.1	0.016 *	0.371
				Women	3.4 ± 1.3		
				Year 3	3.0 ± 1.3		
				Year 4	3.2 ± 1.3		
				Year 5	3.5 ± 1.1		
Unresolved issues around the exams	22.6%	17.8%	59.6%	Men	2.5 ± 1.1	0.000 *	0.226
				Women	3.9 ± 1.1		
				Year 3	3.9 ± 1.3		
				Year 4	3.6 ± 1.3		
				Year 5	3.3 ± 1.1		
Patients were late, did not come or cancelled an appointment shortly before class	17.5%	23.1%	59.4%	Men	3.2 ± 1.0	0.069	0.083
				Women	3.8 ± 1.2		
				Year 3	3.5 ± 1.3		
				Year 4	4.0 ± 1.1		
				Year 5	3.4 ± 1.0		
Not being a good enough dentist when I graduate	17.9%	27.3%	54.8%	Men	3.3 ± 1.2	0.170	0.002 **
				Women	3.8 ± 1.2		
				Year 3	3.3 ± 1.3		
				Year 4	4.2 ± 1.0		
				Year 5	3.3 ± 1.1		
Category «Stressors/infection»				Men	7.5 ± 2.1	0.008 *	0.851
				Women	9.1 ± 2.2		
				Year 3	8.7 ± 2.0		
				Year 4	9.0 ± 2.8		
				Year 5	8.6 ± 2.2		
About being infected with COVID-19 during patient care	40.5%	7.1%	52.4%	Men	2.5 ± 1.6	0.137	0.808
				Women	3.1 ± 1.6		
				Year 3	3.1 ± 1.7		
				Year 4	2.9 ± 1.6		
				Year 5	3.0 ± 1.6		
Ending up in quarantine or home isolation during the pandemic	42.9%	41.6%	15.5%	Men	2.4 ± 1.4	0.055	0.547
				Women	2.9 ± 1.0		
				Year 3	2.8 ± 1.1		
				Year 4	2.9 ± 1.2		
				Year 5	2.6 ± 0.9		
Constant changes in restrictions and infection control routines at the department	40.5%	31%	28.5%	Men	2.7 ± 1.2	0.112	0.700
				Women	3.2 ± 1.2		
				Year 3	3.0 ± 1.0		
				Year 4	3.2 ± 1.2		
				Year 5	3.0 ± 1.2		
Category «Stressors/consequences»				Men	8.9 ± 3.5	0.540	0.335
				Women	9.5 ± 2.8		
				Year 3	9.2 ± 2.6		
				Year 4	9.9 ± 3.1		
				Year 5	8.7 ± 2.9		
The study situation during the pandemic that it exceeded my concentration	28.6%	27.4%	44.0%	Men	2.6 ± 1.2	0.019 *	0.106
				Women	3.3 ± 1.1		
				Year 3	3.3 ± 0.9		
				Year 4	3.4 ± 1.3		
				Year 5	2.6 ± 1.1		
The study situation during the pandemic that it has affected my mental health	38.1%	32.1%	29.8%	Men	2.6 ± 1.6	0.325	0.489
				Women	2.9 ± 1.2		
				Year 3	2.9 ± 1.1		
				Year 4	3.0 ± 1.2		
				Year 5	2.6 ± 1.4		
I have thrived worse in the study compared with before the pandemic	28.6%	25.0%	46.4%	Men	3.7 ± 1.4	0.112	0.483
				Women	3.1 ± 1.3		
				Year 3	3.0 ± 1.5		
				Year 4	3.4 ± 1.2		
				Year 5	3.3 ± 1.3		

* *t*-test. (*p* < 0.05); ** One-way ANOVA-test. (*p* < 0.05).

**Table 3 ijerph-19-03102-t003:** Learning experience.

	Answers in Likert Scale			Mean ± SD		*p*-Value Gender	*p*-Value Year of Study
	1 + 2	3	4 + 5				
How much time have you spent studying since March 2020?	41.7%	33.3%	25.0%	Men	2.6 ± 0.8	0.336	0.568
				Women	2.9 ± 1.0		
				Year 3	2.8 ± 0.9		
				Year 4	3.0 ± 1.0		
				Year 5	2.8 ± 0.9		
How do you assess your theoretical learning outcomes during the pandemic?	59.5%	25.0%	15.5%	Men	2.4 ± 0.7	0.743	0.483
				Women	2.5 ± 0.9		
				Year 3	2.5 ± 1.0		
				Year 4	2.6 ± 0.9		
				Year 5	2.3 ± 0.7		
Year 4 and 5: How do you assess your learning outcome in the clinic/skills course during the pandemic?	59.9%	38.8%	1.3%	Men	1.9 ± 0.7	0.027 *	0.473
				Women	2.4 ± 0.7		
				Year 4	2.3 ± 0.8		
				Year 5	2.4 ± 0.6		
Year 3: How do you assess your learning outcome in the clinic/skills course during the pandemic?	26.9%	42.3%	30.8%	Men	3.2 ± 0.8	0.613	-
				Women	3.0 ± 0.8		
				Year 3	3.0 ± 0.8		
Has there been good enough facilitation from the department to compensate for lost clinic time/skills training?	32.1%	45.2%	22.7%	Men	3.1 ± 1.0	0.344	0.359
				Women	2.8 ± 1.0		
				Year 3	2.8 ± 0.8		
				Year 4	2.7 ± 0.9		
				Year 5	3.1 ± 1.0		

* *t*-test. (*p* < 0.05).

## Data Availability

Data are available on request from the authors.

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
