# Peer review of "Dental Students’ Experiences during the COVID-19 Pandemic—A Cross-Sectional Study from Norway"

_ijerph, 2022, doi:10.3390/ijerph19053102_

Round 1

Reviewer 1 Report

Dear Authors,

The article: 'Dental students’ experiences during COVID-19 pandemic – a cross-sectional study from Norway' wasto map dental students' experience of the study situation throughout the pandemic.

English language and style must be corrected.

Punctuation mistakes should be corrected. 

Add this article in introduction: https://doi.org/10.3390/jcm9103344

p value is written in italics.

Tables should be prepared using MDPI guidelines.

In table 2 and 3, column 5 should be wider than the others.

Figures 1 and 2 - remove the title of the figure in the figure, if it is in the caption.

Add a table with abbreviations.

References should be prepared in accordance with the MDPI guidelines

Add the research form in the additional materials. 

To sum up, article should be reconsider after minor revision.

Author Response

Thank you very much for your in-depth revision of our manuscript. The revisions in the manuscript are marked using the “Track Changes” function in MS Word as suggested. This is used on revisions to add new information in line with the suggestions from the reviewers. Additionally, new information is marked with yellow to clarify these changes since revision was done to enhance the quality of the language and content.

The article: 'Dental students’ experiences during COVID-19 pandemic – a cross-sectional study from Norway' was to map dental students' experience of the study situation throughout the pandemic.

English language and style must be corrected. - Language and style were corrected.

Punctuation mistakes should be corrected. – Punctuation was corrected.

Add this article in introduction: https://doi.org/10.3390/jcm9103344  - The article was added in the introduction as suggested.

p value is written in italics. -  It was changed as suggested.

Tables should be prepared using MDPI guidelines. - It was corrected accordingly to the MDPI guidelines.

In table 2 and 3, column 5 should be wider than the others. - It was done as suggested.

Figures 1 and 2 - remove the title of the figure in the figure, if it is in the caption. - It was done as suggested.

Add a table with abbreviations.- Abbreviations were explained in the text, and we changed the abbreviations in the tables into words.

References should be prepared in accordance with the MDPI guidelines - References were corrected according to MDPI guidelines

Add the research form in the additional materials. – The questionnaire was made in Norwegian, and we have not done a lingvistic validation. Therefore, we didn’t upload the questionnaire in the additional materials. If someone is interested, we can share the Norwegian version.

To sum up, article should be reconsidered after minor revision. Thank you for the very in-depth and valuable revision

Reviewer 2 Report

"physical education" change with hands-on training 

"Due to these stressors and demands, dental students are prone to devel
oping mental health problems" very strong affirmation...maybe too much strong... add some references

Have You checked the normal distribution before to use T-STUDENT and ANOVA?

I strongly recommend a post-hoc analysis to better understand the differences between the categories

In the discussion I would face firstly how generally medical and healthcare students and not only dental students face anxiety and stress  during their studies 

https://www.mdpi.com/2076-3417/10/7/2357; https://www.mdpi.com/1660-4601/16/15/2735

As You put the theme of difference between gender manuscript lacks of article not finding these type of differences  and the discussion results quite sexist. 

I suggest to discuss Your data and the true aim of Your study with similar studies carried out

https://bmcmededuc.biomedcentral.com/articles/10.1186/s12909-020-02257-4

https://www.mdpi.com/2227-9032/9/4/454

https://mededu.jmir.org/2021/2/e25506/

https://bmjopen.bmj.com/content/11/12/e054728.long

Gender data is interesting but it is not appropriately discussed and really does not give a nice impression

Most students wanted to return to physical lectures after the pandemic 

replace physical with "in presence" 

Author Response

The revisions in the manuscript are marked using the “Track Changes” function in MS Word as suggested. This is used on revisions to add new information in line with the suggestions from the reviewers. Additionally, new information is marked with yellow to clarify these changes since revision was done to enhance the quality of the language and content.

physical education" change with hands-on training.

 - We did not mean “hands-on training” when we wrote “physical education” in the introduction. We meant all traditional teaching, including both theoretical and clinical education. Thank you for the comment, so that we could clarify this. We made this change: “Universities, among other institutions and businesses, were closed down, and physical education all traditional teaching had to be quickly converted into digital education.”

"Due to these stressors and demands, dental students are prone to developing mental health problems" very strong affirmation...maybe too much strong... add some references – The sentence was rechanged and reference has been added.

Have You checked the normal distribution before to use T-STUDENT and ANOVA?

 – The Likert scale is evidently not normal distributed, even if it may be equidistant. However, when inspecting the residuals after testing the mean values, we found that these were bell shaped. Furthermore, due to the relatively large sample size, the t-test and ANOVA will be robust tests for the mean values. Hence, we considered these tests to be appropriate.

I strongly recommend a post-hoc analysis to better understand the differences between the categories.

- For the analysis, we have only presented the overall p-values (for “gender” and “year of study”). For “year of study” we could have used a linear model, setting one of the years as a reference, comparing the two others to this reference, and then perform post-hoc analysis adjusted for multiple comparisons (e.g. using Scheffé’s adjustment). This would generate 3 times as many p-values for “year of study” as is now presented. This will, in our opinion complicate the presentation of the results excessively. Moreover, we wanted to focus on the students’ overall experience, not the genders or years of study in particular.

In the discussion I would face firstly how generally medical and healthcare students and not only dental students face anxiety and stress during their studies  - https://www.mdpi.com/2076-3417/10/7/2357 ,https://www.mdpi.com/1660-4601/16/15/2735

- We added a paragraph about how medical and healthcare students faced anxiety and stress, and we cited suggested articles.

As You put the theme of difference between gender manuscript lacks of article not finding these type of differences  and the discussion results quite sexist.

 – We have found more articles to discuss these differences in the discussion section. Moreover, we wanted to focus on the students’ overall experience, not the genders or years of study in particular.

I suggest to discuss Your data and the true aim of Your study with similar studies carried out https://bmcmededuc.biomedcentral.com/articles/10.1186/s12909-020-02257-4

https://www.mdpi.com/2227-9032/9/4/454

https://mededu.jmir.org/2021/2/e25506/

https://bmjopen.bmj.com/content/11/12/e054728.long

- The aim of our study was re-shaped, and we included all the suggested articles in our discussion section and discussed them with our data. Thank you for very good articles that improved the quality of our article.

Gender data is interesting, but it is not appropriately discussed and really does not give a nice impression

- We have found more articles to discuss this in the discussion section. Moreover, we wanted to focus on the students’ overall experience, not the genders or years of study in particular.

Most students wanted to return to physical lectures after the pandemic, replace physical with "in presence" – It was replaced as suggested.

Reviewer 3 Report

Thank you for the opportunity to review your research. It tells quite interestingly about the current situation in your countries.

However, in order to increase scientific significance, it is necessary to use more valid, standardized methods and it is imperative to correlate empirical data with facts that reflect, for example, academic performance.

First, Likert scales are arbitrary. The value assigned to each Likert item is simply determined by the researcher designing the survey, who makes the decision based on a desired level of detail. Thus, the range captures the intensity of their feelings for a given situation.

The authors say they did a stress study, but line 92 says "The questionnaire was designed specifically for this study". Still, the assessment of psycho-emotional stress should be carried out according to valid and generally accepted methods, of which there are quite a lot: these can be various tests such as Sheehan, Spielberger, State-Trait Anxiety Inventory (STAI), based on a 4-point Likert scale. In addition to assessing the psycho-emotional state, some kind of verification of the reliability of this relationship is needed, for example, the results of an intermediate certification, as was done in this study DOI: 10.28991/esj-2021-SPER-07, doi: 10.2196/22817,  doi.org/10.1111/ppc.12597, the design of which is similar to the current one.

The concept of stress for the layman has a different degree of severity and it is necessary to offer to evaluate its degree using more sensitive methods. So graph 1 (When did you feel the most stressed?) shows the subjective attitude of the study participants, the interpretation of which can only be at the level of informing.

Tell me, were there psychologists among the authors who helped in the development of rating scales and interpretation of the results?
In this regard, I would like to see the use of standardized methods for assessing stress, as well as evaluating the results in terms of student achievement. This will allow us to assert that there is a connection between the obtained results.

Author Response

Thank you very much for your in-depth revision. The revisions in the manuscript are marked using the “Track Changes” function in MS Word as suggested. This is used on revisions to add new information in line with the suggestions from the reviewers. Additionally, new information is marked with yellow to clarify these changes since revision was done to enhance the quality of the language and content.

Thank you for the opportunity to review your research. It tells quite interestingly about the current situation in your countries.

However, in order to increase scientific significance, it is necessary to use more valid, standardized methods and it is imperative to correlate empirical data with facts that reflect, for example, academic performance.

First, Likert scales are arbitrary. The value assigned to each Likert item is simply determined by the researcher designing the survey, who makes the decision based on a desired level of detail. Thus, the range captures the intensity of their feelings for a given situation.

The authors say they did a stress study, but line 92 says "The questionnaire was designed specifically for this study". Still, the assessment of psycho-emotional stress should be carried out according to valid and generally accepted methods, of which there are quite a lot: these can be various tests such as Sheehan, Spielberger, State-Trait Anxiety Inventory (STAI), based on a 4-point Likert scale. In addition to assessing the psycho-emotional state, some kind of verification of the reliability of this relationship is needed, for example, the results of an intermediate certification, as was done in this study DOI: 10.28991/esj-2021-SPER-07, doi: 10.2196/22817,  doi.org/10.1111/ppc.12597, the design of which is similar to the current one.

– We agree that assessment of psycho-emotional stress should be carried out according to valid and generally accepted methods. However, our aim was to present stressors that dental students had experienced in the study situation during the pandemic, not overall stress levels. We see that this did not appear clearly in the manuscript, thank you very much for correcting us. In the research for this study, we considered using different tools to measure stress levels than: the Depression, Anxiety and Stress Scale (DASS) (Lovibond, Lovibond) and the aforementioned State-Trait Anxiety Inventory (STAI). In our opinion these questionnaires did not fit into the aim of our study. “STAI is used to measure trait and state anxiety. It can be used in clinical settings to diagnose anxiety and to distinguish it from depressive syndromes.” (Spielberger, Gorsuch, Lushene, Vagg, & Jacobs, 1983). The pandemic influenced the entire society and that is why we have decided to present stressors that dental students were facing during the pandemic. Thank you for correcting us. We have changed in text that we focused on stressors and not on stress.

The concept of stress for the layman has a different degree of severity and it is necessary to offer to evaluate its degree using more sensitive methods. So graph 1 (When did you feel the most stressed?) shows the subjective attitude of the study participants, the interpretation of which can only be at the level of informing.- We agree and have removed this graph.

Tell me, were there psychologists among the authors who helped in the development of rating scales and interpretation of the results?In this regard, I would like to see the use of standardized methods for assessing stress, as well as evaluating the results in terms of student achievement. This will allow us to assert that there is a connection between the obtained results.

-  There was no psychologist among us. This was one of the reasons why we concentrated on stressors that dental students had faced during the pandemic, and not overall stress levels. We have corrected in the manuscript that we focus on stressors. Additionally, in the limitation of the study there is information that “…the findings are unique only for this sample and cannot be generalized to all dental schools and dental educations.”

Moreover, it was difficult to evaluate the student achievement in terms of e.g. grades, because the exams were changed during the pandemic. Instead, we looked at the learning experiences among the students (table 3) and consequences (quitting/applying for leave due to the pandemic).

Round 2

Reviewer 1 Report

Article can be accepted after editor decison.

Reviewer 2 Report

I am very well impressed by the authors work and replies. 
congratulations. To me manuscript can be now accepted 

Reviewer 3 Report

Hello dear authors!

Thank you for the corrections. In its present form, your research fully satisfies me and you answered the questions that were posed